# Effects of Second-Season Crops on Soybean Cultivation in Compacted Soil in Brazilian *Cerrado*

Camila Jorge Bernabé Ferreira [1,*], Alessandro Guerra da Silva [1], Vívian Ribeiro de Oliveira Preto [1], Cássio Antonio Tormena [2], Guilherme Braga Pereira Braz [1], Matheus de Freitas Souza [1] and André Luiz Biscaia Ribeiro da Silva [3]

[1] Postgraduate Program in Plant Production, Universidade de Rio Verde, Rio Verde 75901-970, Brazil
[2] Department of Agronomy, Universidade Estadual de Maringá, Maringa 87020-900, Brazil
[3] Department of Horticulture, Auburn University, Auburn, TX 36849, USA
* Correspondence: camilajbferreira@gmail.com; Tel.: +55-064-3611-2200

**Abstract:** In no-tillage systems, soil compaction has caused negative impacts on crop productivity and soil quality. The objective of this study was to evaluate the soil physical quality after different crops of the second season: maize, sorghum, and millet in compacted and uncompacted soils, in addition to evaluating the performance of soybean in succession in Rhodic Ferralsol under no-tillage (NT) in the Brazilian *Cerrado* biome. A field experiment was conducted during the second season of 2019 and the first season of 2019/20 in Rio Verde, Brazil. The experimental design used randomized blocks in a $3 \times 2$ factorial scheme, with six replications. The first factor corresponded to the cultivation of maize, sorghum, and millet; the second factor was the cultivation of these crops in compacted and uncompacted soils. The physical properties of the soils in the 0–0.1 and 0.1–0.2 m depth layers were evaluated after the second season of cultivation, in addition to the agronomic characteristics of the soybean cultivated in succession. The results indicate that the compaction influenced the soil physical quality, mainly in the 0.1–0.2 m layer, reflecting a decrease in the performance of the soybean crop (i.e., the plant height, number of pods per plant, and grain yield). The use of the second-season crop of millet improved the soil physical properties of penetration resistance and macroporosity and improved the water/air relation. The use of millet provided a reduction of up to 20% in the soil penetration resistance. About 10% more soybean was produced after cultivation in succession to millet compared to maize and sorghum.

**Keywords:** *Glycine max*; *Pennisetum glaucum*; soil penetration resistance; *Sorghum bicolor*; *Zea mays*

## 1. Introduction

Due to adequate climatic and edaphic conditions, the Central-West region of Brazil is the one that most contributes to the country's grain production, which comprises a large part of the *Cerrado* biome [1]. The main cropping system adopted for grain production in the region is no-tillage (NT), consisting of minimal soil disturbance, the use of straw residuals, and crop rotation [2]. In particular, this cropping system has been widely adopted in the region, but only soybean (*Glycine max*) and maize (*Zea mays*) are rotated, with both crops being planted in a single season. Soybean is planted in early summer, and maize is planted right after the soybean harvest [3]. Because the two cash crops are planted in the same year, the approach attracts growers due to its potential profitability [4]. However, the negative effects of agricultural intensification on cropland have already been described by Barnerjee et al. and Anghinoni et al. [5,6]. These authors reported that poor crop diversification in production systems negatively impacts soil physical quality, reduces soil organic matter content, and decreases crop productivity [5,6].

Regarding soil physical quality, the soil compaction in NT agricultural systems has increased, which is caused by the intense traffic of heavy machinery and inadequate

management practices [7]. Soil compaction is the process by which soil solids are rearranged to decrease void space and bring them into closer contact with each other, increasing soil bulk density [8]. As a consequence of compaction, there is a reduction in the growth of plant roots, caused by an increase in mechanical strength and a decrease in soil aeration, that ultimately reduces crop yields [9].

Currently, climate projections indicate an increase in the frequency and severity of drought [10]. With this scenario, the harmful effects of soil compaction on cropping systems can be intensified on crops. [11]. In compacted soils, it is important to seek alternatives to minimize crop yield losses. Maize, the main crop in succession to soybeans in the Center-West region of Brazil, may have its development impaired due to soil compaction, as it is a crop that is not tolerant to prolonged periods of water deficit, in addition to being sensitive to soil compaction [12]. As an alternative, areas with sorghum (*Sorghum bicolor*) and millet (*Pennisetum glaucum*) cultivation have lower water requirements as a way of increasing water use efficiency. In addition, millet has the advantage of presenting an aggressive root system with the potential to mitigate soil compaction [13].

The hypothesis established in this study is that the use of alternative crops as second crops can mitigate soil compaction and ensure a greater yield of soybean grown in succession. The objective of this study was to evaluate the use of second-season crops (i.e., maize, sorghum, and millet) in compacted and uncompacted soils and their influence on soil physical quality and the performance of soybeans cultivated in succession under NT in a Rhodic Ferralsol in the *Cerrado* Biome in the Central-West region of Brazil.

## 2. Materials and Methods

### 2.1. Area Characterization and Experimental Design

The field experiment was conducted in the 2019/20 season in Rio Verde, Goias (17°46′52.03″ S; 50°58′13.46″ W; and an altitude of 789 m) in the Center-West region of Brazil. The experimental area was farmed using NT for more than 10 years, with soybeans being cultivated in the spring/summer seasons and maize in succession in the second or autumn/winter seasons.

The dominant climate in the region is characterized as the Aw type (a tropical climate with a defined dry season) according to the Köppen classification [14]. The soil is classified as Rhodic Ferralsol according to [15] or as Latossolo Vermelho distrófico according to the Brazilian classification system [16] due to its clay loam texture. The average air temperature and precipitation data during the experiment are shown in Figure 1.

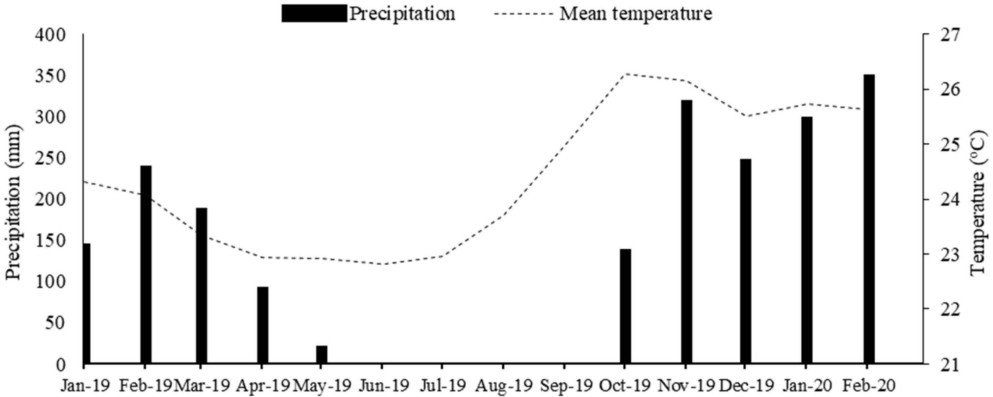

**Figure 1.** Average air temperature and precipitation values during the experimental period. Rio Verde, Goias, 2019/2020. Source: National Institute of Meteorology (INMET)—Climatological Station of the University of Rio Verde.

Prior to the installation of the experiment, a soil sample was carried out in the 0.0–0.2 m soil depth layer, and the soil physicochemical properties were characterized, such as pH(CaCl$_2$) 5.10; 0 cmol$_c$ dm$^{-3}$ of Al$^{+3}$; 4.28 cmol$_c$ dm$^{-3}$ of Ca$^{+2}$; 2.09 cmol$_c$ dm$^{-3}$ of Mg$^{+2}$;

0.59 $cmol_c$ $dm^{-3}$ of $K^+$; 16.08 mg $dm^{-3}$ of available P (Mehlich-1); 18.35 g $dm^{-3}$ of organic carbon (Walkley-Black); and 495, 50, and 455 g $kg^{-1}$ of sand, silt, and clay, respectively. The mineralogical composition of the soil was iron and aluminum oxides [17]. Subsequently, a two-factor experimental design of a second-season crop and soil compaction was arranged in a randomized complete block design with six replications. The second-season crop treatments consisted of maize, grain sorghum, and millet. When planting maize, sorghum, and millet, the following materials and sowing densities were used, respectively: P3646YHR® (Pioneer Seeds, Santa Cruz do Sul, Brazil) at a density of 3 seeds $m^{-1}$; BRS 380® (EMBRAPA, Brasília, Brazil) at a density of 10 seeds $m^{-1}$); and ADR 300® (ATTO Seeds, Rondonópolis, Brazil) at a density of 9 plants $m^{-1}$. The soil compaction treatments consisted of the presence or absence of soil compaction. Each experimental unit comprised eight planting rows that were 5.0 m in length and spaced at 0.45 m (18.0 $m^2$).

The soil compaction treatments were applied on 25 October 2018, in which compaction was applied after a 30 mm rain with a tractor. The tractor was a John Deere, model 5403, with a total mass of 4510 kg, diagonal tires, front/rear tire measurements of 14.9/28″ and 23.1/30″, and inflation pressures of 85 and 60 kPa, respectively. After soil compaction implementation, an evaluation of penetration resistance (PR) was carried out using a Falker® brand penetrometer, model PLG 1020 (FALKER, Porto Alegre, Brazil), in which the PR data were measured every 0.01 m to a 0.40 m soil depth. Measurements were taken two days after the occurrence of rain, so the soil water content was close to the field capacity (measured moisture = 0.23 ± 0.03 kg $kg^{-1}$). After measuring PR, the following results were obtained: compacted soil at depths of 0–0.1, 0.1–0.2, 0.2–0.3, and 0.3–0.4 (m), with values of PR 4.01 ± 0.22 (mean ± standard deviation), 3.41 ± 0.20, 3.58 ± 0.25, and 4.08 ± 0.31 MPa, respectively; while for the uncompacted soil, the PR values were 1.98 ± 0.30, 2.67 ± 0.21, 2.88 ± 0.19, and 2.77 ± 0.22 MPa, for the respective sampled depths.

Second-crop treatments were planted on 28 February 2019 with the application of 170 kg $ha^{-1}$ of monoammonium phosphate. In topdressing, 150 kg $ha^{-1}$ of nitrogen was applied 15 days after the emergence (DAE) of each crop, using urea as a source. The cultural treatments were made according to the needs of each crop. The yields obtained for each crop are presented in Table 1.

**Table 1.** Yields of maize, sorghum, and millet in compacted and uncompacted soils.

| Crop | Compacted | Uncompacted |
|------|-----------|-------------|
| Grain yield—Maize (kg $ha^{-1}$) | 5413 ± 242 [1/] | 6143 ± 183 |
| Grain yield—Sorghum (kg $ha^{-1}$) | 4676 ± 224 | 5231 ± 287 |
| Dry mass yield—Millet (kg $ha^{-1}$) | 6852 ± 422 | 8778 ± 325 |

[1/] Standard deviation.

Soybean was planted on 11 November 2019 with the soybean cultivar FLX IPRO® (BRASMAX Genética, Cambe, Brazil) at a plant density of 17 seeds $m^{-1}$. This cultivar has the characteristic of being early-cycle (maturation group 6.6), of indeterminate growth habit, and high yield potential. Seeds were treated, and an inoculant was applied, according to the cultivar recommendations. Monoammonium phosphate (MAP) (138 kg $ha^{-1}$) was used at the time of sowing, and 120 kg $ha^{-1}$ of potassium chloride (KCl) was used when the crop was at phenological stage V3. The emergence of soybean seedlings occurred on 16 November 2019. During the development of the soybean crop, all cultural practices were carried out according to the recommendations, proceeding to control pests, diseases, and weeds without letting them influence the development of the crop.

### 2.2. Sampling and Soil Physical Determinations

Soil samples were collected in September 2019 (before soybean sowing) at 0.0 to 0.1 and 0.1 to 0.2 m soil depths. Small trenches (0.25 × 0.25 m) were opened, and two undisturbed samples were collected per experimental unit and layer using stainless-steel rings 0.05 m in diameter and 0.05 m in height (volume of $10^{-4}$ $m^3$), totaling 144 samples. The soil samples

were wrapped in aluminum foil, carefully transported to the laboratory, and kept at 4 °C to avoid interference with biological activity. Subsequently, the samples were placed on trays and saturated with water by capillarity for 48 h. Once saturated, the samples were weighed and subjected to a potential of −6 kPa using a tension table similar to that described by Ball and Hunter [18]. Upon reaching equilibrium, as indicated by the absence of water draining from the tension table, the samples were weighed again and dried in an oven at ±105 °C for 24 h. The soil bulk density (BD) was calculated as the ratio between the dry mass of the soil and the total volume of the sample, as described by Grossman and Reinsch [19].

The total porosity (TP) was determined by the following equation, where TP: total porosity ($m^3\ m^{-3}$); BD: soil bulk density ($Mg\ m^{-3}$); and Dp: particle density, which was obtained in a sample by repetition according to the methodology described in Embrapa [20], with an average value of $2.60 \pm 0.03\ Mg\ m^{-3}$.

$$TP = 1 - (BD/Dp)$$

The water content in the field capacity (FC) was obtained indirectly by the water content of the soil retained in the matrix with a potential of −6 kPa, according to Severiano et al. [21]. The soil aeration porosity (SAC), or the volume of drained pores after the equilibrium of the soil in the sample with a potential of −6 kPa, was determined by the difference between the total porosity and the water content with the potential of −6 kPa. The soil water storage capacity indicator (FCTP, dimensionless) was calculated as the ratio between the water content at the potential of −6 kPa (taken as the water content at the field capacity) and the total porosity of the soil. The soil air storage capacity indicator (SACTP, dimensionless) was calculated as the ratio between the soil porosity with air at the potential of −6 kPa and the total porosity [22].

Simultaneous to soil sampling with the preserved structure, measurements of soil PR were carried out using a Falker® penetrometer. Measurements were carried out two days after sufficient rain to moisten the sampled profile, so the soil water content was close to the field capacity. The water content was $0.24 \pm 0.02\ kg\ kg^{-1}$. Four PR subrepetitions were performed in each repetition.

### 2.3. Evaluations of the Soybean Crop

After planting the soybean crop, the plant population was evaluated at 20 days after emergence (DAE). At harvest, three random plants per plot were evaluated for plant height (measurement from the collar to the insertion of the last trifoliate), the total number of pods per plant (counting the number of pods on the main and secondary stems), the thousand-grain weight (by counting and weighing a thousand grains, with moisture correction to 13%), and yield (by weighing the grains of the harvested plants, with subsequent moisture correction to 13%).

### 2.4. Statistical Analysis

An analysis of variance was performed for all soil and plant parameters using SAS [23]. When the significance of the treatments was verified, the LSD-Fisher Test was used ($p \leq 0.05$). A principal component analysis (PCA) was performed to determine correlations among the crop variables and responses using R software (version 4.4.6, factorminer and factorextra packages).

## 3. Results

Regardless of the soil compaction treatment, the PR measured in the field was influenced by the second-season crops (Figure 2). The PR reduced along the soil profile when millet was used compared to maize and sorghum. This result may be due to the effect of the millet root system, which has voluminous and deep roots that are capable of reducing soil compaction [13]. On the soil surface (0–0.05 m), the PR values above the critical limit of 2.5 MPa [24] were only measured in the soil compacted condition with maize cultivation. At the other depths (0.05–0.35 m), all crops presented values higher than that considered

impeditive to root growth, regardless of the soil condition. However, the lowest values of PR were reached with the use of millet, which was efficient in attenuating up to 20% the magnitude of this variable compared to maize and sorghum. In particular, the use of millet was efficient in attenuating soil compaction and providing a greater volume of soil with less restrictive penetration resistance, which allows deeper rooting and increased access to water and nutrients by plants.

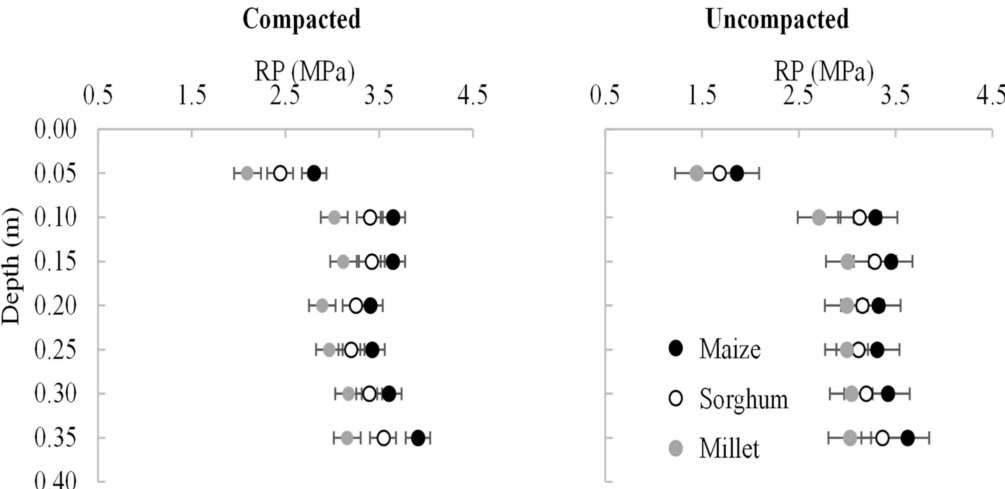

**Figure 2.** Soil penetration resistance measured in the field after second-season crops of maize, sorghum, and millet in compacted and uncompacted soils. The bars refer to the confidence interval of the mean, and the overlap of the confidence intervals indicates the absence of differences between means ($p \leq 0.05$).

In the 0–0.1 m soil layer, there were no significant differences between the soil compaction treatments for soil microporosity, macroporosity, water storage capacity, air storage capacity, and soil porosity (Table 2). However, the soil bulk density was higher in the compacted soil treatment compared to the uncompacted treatment, as expected. For the comparison made among the second-crop treatments, there was greater macroporosity and air storage capacity in the 0–0.1 m soil layer when millet was cultivated as a second-season crop. In soils cultivated with maize and sorghum, the soil water storage capacity considerably exceeded the limit of 0.66 proposed by Reynolds et al. [21] as an ideal condition, showing values above 0.88. Contrarily, the indices of the soil water and air storage capacity in soils cultivated with millet were the closest to the ideal conditions of 0.66 and 0.33, respectively.

In the 0.1–0.2 m soil layer, the highest soil bulk density and soil water storage capacity and the lowest macroporosity and air storage capacity were measured in the compacted soil treatment, demonstrating the residual effect of soil compaction in this layer, promoting an unfavorable physical environment for root growth [25].

There was a significant interaction between the soil compaction and second-crop treatments for the soil bulk density and total porosity in the soil layer of 0.1–0.2 m. In particular, there was no significant difference between the soil compaction treatments within the millet treatment for the soil bulk density and total porosity, which can be associated with the growth of the millet root system, which acted significantly in this layer. For the soil macroporosity, soil air storage capacity, and soil water storage capacity, there was no difference between maize and millet cultivation in the 0.1–0.2 m layer.

In addition to measuring the response variables related to soil physical properties, evaluations were carried out to assess the impact of improvements in the edaphic environment on soybeans planted in succession to crops used as second-season treatments (Table 3). However, the soybean plant population had no influence from the soil compaction treatment or the second-crop treatments (maize, sorghum, and millet), with an average of 405.5 thousand plants ha$^{-1}$.

**Table 2.** Mean values of soil physical properties (field capacity—FC; soil aeration capacity—SAC; soil water storage capacity—FCTP; air storage capacity—SACTP; total porosity—TP; soil bulk density—BD) in the 0.0 to 0.1 and 0.1 to 0.2 m soil layers after second-season crops of maize, sorghum, and millet in compacted (C) and uncompacted (NC) soils.

| | Crops | 0.0–0.1 m | | | 0.1–0.2 m | | |
|---|---|---|---|---|---|---|---|
| | | C | NC | Mean | C | NC | Mean |
| BD (Mg m$^{-3}$) | Maize | 1.37 Aa [1/] | 1.30 Aa | 1.33 a | 1.54 Ba | 1.46 Aa | 1.51 a |
| | Sorghum | 1.36 Aa | 1.34 Aa | 1.35 a | 1.55 Ba | 1.48 Aa | 1.51 a |
| | Millet | 1.34 Aa | 1.31 Aa | 1.32 a | 1.51 Aa | 1.49 Aa | 1.50 a |
| | Mean | 1.36 B | 1.32 A | | 1.54 B | 1.48 A | |
| Microporosity (m$^3$ m$^{-3}$) | Maize | 0.49 Aa | 0.51 Aa | 0.50 a | 0.35 Ab | 0.35 Ab | 0.35 b |
| | Sorghum | 0.47 Ab | 0.49 Aa | 0.48 b | 0.38 Aa | 0.38 Aa | 0.38 a |
| | Millet | 0.40 Ac | 0.41 Ab | 0.41 c | 0.32 Ac | 0.33 Ab | 0.33 c |
| | Mean | 0.45 A | 0.47 A | | 0.35 A | 0.35 A | |
| Macroporosity (m$^3$ m$^{-3}$) | Maize | 0.05 Ab | 0.06 Ab | 0.06 b | 0.12 Aa | 0.15 Aa | 0.14 a |
| | Sorghum | 0.05 Ab | 0.06 Ab | 0.05 b | 0.08 Aa | 0.10 Aa | 0.09 b |
| | Millet | 0.14 Aa | 0.15 Aa | 0.14 a | 0.14 Aa | 0.14 Aa | 0.14 a |
| | Mean | 0.08 A | 0.09 A | | 0.11 B | 0.13 A | |
| FCTP | Maize | 0.89 Ab | 0.90 Ab | 0.90 b | 0.74 Ab | 0.70 Aa | 0.72 a |
| | Sorghum | 0.88 Ab | 0.91 Ab | 0.90 b | 0.82 Ac | 0.79 Ab | 0.80 b |
| | Millet | 0.74 Aa | 0.74 Aa | 0.74 a | 0.69 Aa | 0.70 Aa | 0.70 a |
| | Mean | 0.84 A | 0.85 A | | 0.76 B | 0.73 A | |
| SACTP | Maize | 0.10 Ab | 0.10 Ab | 0.10 b | 0.25 Ba | 0.30 Aa | 0.28 a |
| | Sorghum | 0.12 Ab | 0.08 Ab | 0.10 b | 0.18 Ab | 0.21 Ab | 0.19 b |
| | Millet | 0.26 Aa | 0.26 Aa | 0.26 a | 0.30 Aa | 0.30 Aa | 0.30 a |
| | Mean | 0.16 A | 0.15 A | | 0.24 B | 0.27 A | |
| TP (m$^3$ m$^{-3}$) | Maize | 0.55 Aa | 0.57 Aa | 0.56 a | 0.46 Ba | 0.50 Aa | 0.48 a |
| | Sorghum | 0.53 Aa | 0.54 Aa | 0.54 a | 0.45 Ba | 0.47 Aa | 0.46 a |
| | Millet | 0.56 Aa | 0.55 Aa | 0.55 a | 0.48 Aa | 0.48 Aa | 0.48 a |
| | Mean | 0.54 A | 0.56 A | | 0.47 B | 0.48 A | |

[1/] Means followed by uppercase letters in the row and lowercase letters in the column do not differ from each other by the LSD-Fisher test ($p \leq 0.05$).

**Table 3.** Agronomic characteristics (plant population, plant height, number of pods per plant, and thousand-grain weight) of soybean cultivated after second-season crops of maize, sorghum, and millet in compacted (C) and uncompacted (NC) soil.

| Crops | C | NC | Mean | C | NC | Mean |
|---|---|---|---|---|---|---|
| | Population (Thousand Plants ha$^{-1}$) | | | Plant Height (cm) | | |
| Maize | 399.9 Aa [1/] | 411.1 Aa | 405.5 a | 64.7 bB | 71.9 bA | 68.3 b |
| Sorghum | 407.3 Aa | 385.1 Aa | 396.2 a | 62.8 bB | 68.2 bA | 65.5 b |
| Millet | 405.5 Aa | 423.9 Aa | 414.8 a | 76.3 aA | 79.0 aA | 77.7 a |
| Mean | 404.3 A | 406.74 A | | 68.0 B | 73.0 A | |
| | Number of pods | | | Thousand-grain weight (g) | | |
| Maize | 20.3 Bb | 34.3 Aa | 27.3 b | 184.4 Aa | 184.7 Ab | 184.7 b |
| Sorghum | 19.8 Bb | 29.5 Ab | 24.6 c | 188.8 Aa | 185.8 Ab | 187.3 b |
| Millet | 28.6 Ba | 33.3 Aa | 31.0 a | 190.5 Aa | 196.9 Aa | 193.7 a |
| Mean | 22.9 B | 32.4 A | | 188.0 A | 189.1 A | |

[1/] Means followed by uppercase letters in the row and lowercase letters in the column do not differ from each other by the LSD-Fisher test ($p \leq 0.05$).

For plant height, there was a significant interaction between the soil compaction and second-crop treatments (Table 3). In general, when the soil physical condition was compared among the species exploited in the second season, it appeared that, for maize and sorghum, soybean plants that developed in uncompacted soil presented greater heights, to

the detriment of those that developed in compacted soil. Additionally, regardless of the soil physical condition, soybean plants that were planted after millet had higher plant heights compared to those planted after sorghum and maize.

The number of pods per soybean plant was also significantly affected by the interaction between the soil compaction and second-crop treatments (Table 3). Regardless of the second-crop treatment, plants grown in uncompacted soil had a greater number of pods compared to those grown in compacted soil, with an average increase of ≈41.5% in the total number of pods per plant. Furthermore, another important observation that can be made refers to the fact that in compacted soil the cultivation of millet before soybean provided increases in the number of pods, while in uncompacted soils the values of this variable in soybean plants were similar in treatments that had maize and millet as predecessor crops, with the number of pods in these treatments being higher than those observed after sorghum.

There was also a significant interaction between the soil compaction and second-crop treatments for the thousand-grain weight of soybean; however, the significant differences were only measured within the uncompacted soil, where plants of soybeans that were cultivated after millet had the highest thousand-grain weight (Table 3). For soybean grain yield, there were significant main effects of the soil compaction and second-crop treatments (Table 4).

**Table 4.** Soybean grain yield after second-season crops of maize, sorghum, and millet in compacted (C) and uncompacted (NC) soils.

| Crops | C | NC | Mean |
|---|---|---|---|
| | Grain Yield (kg ha$^{-1}$) | | |
| Maize | 3380.1 Aa [1/] | 3684.1 Aa | 3532.1 b |
| Sorghum | 3318.5 Aa | 3694.8 Aa | 3506.7 b |
| Millet | 3742.7 Aa | 4066.5 Aa | 3904.5 a |
| Mean | 3480.3 B | 3815.2 A | |

[1/] Means followed by uppercase letters in the row and lowercase letters in the column do not differ from each other by the LSD-Fisher test ($p \leq 0.05$).

Regarding the effect of the soil physical condition, it was noted that the condition of compacted soil provided a decrease in crop yield of 335 kg ha$^{-1}$ (9.6%) compared to the uncompacted soil. Regarding the effect of predecessor crops influencing grain yield, it was observed that millet provided an increase in soybean yield compared to sorghum and maize, with no significant differences between these last two plant species. Treatments containing millet cultivation in the second crop prior to soybean sowing provided an average increase of ≈10.6% compared to those with maize or sorghum.

Figure 3a shows the principal component analysis of the agronomic characteristics of soybean with soil physical properties. The two main axes of the PCA explained 98.8% of the total data variance. The macroporosity, soil air storage capacity, and total porosity were the soil physical properties that had positive relationships with plant evaluations, especially the number of pods per plant and height, which had the greatest relationship with the condition of uncompacted soil. The soil water storage capacity and soil bulk density had a greater relationship with the compacted soil condition.

Figure 3b shows principal components 1 and 2, with second-season crops and soil conditions as explanatory variables. The two main axes of the PCA explained 96.8% of the total data variance. PC1 separated Macro and SACTP on the positive side from Micro, FCTP, and BD on the negative side. The data show that PC1 represents the soil condition and second-season crop characteristics that are directly proportional to the MACRO and SACTP contents of millet in compacted and uncompacted soil conditions. This analysis distinguishes the treatment with millet in different groups from the other treatments, as the use of millet in the second season provided a better soil physical environment with a better soil physical quality condition for root growth and plant development than the other systems, independent of the soil conditions.

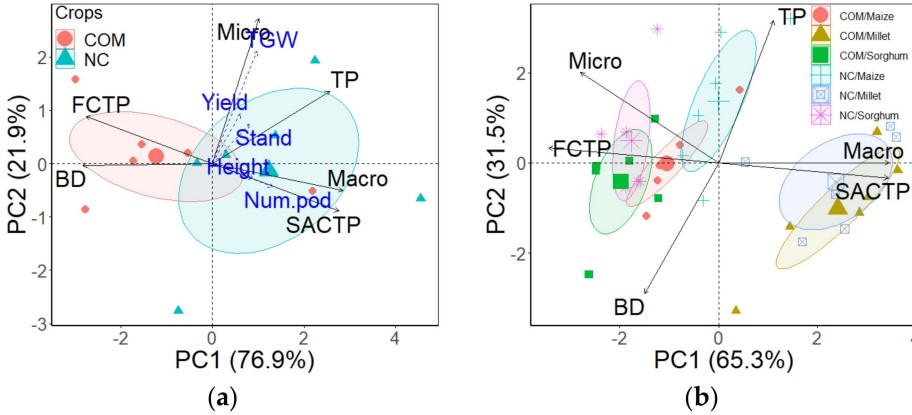

**Figure 3.** Principal component analysis of soil conditions related to soil physical properties and agronomic characteristics (**a**) and second-season crops and soil conditions with soil physical properties (**b**). COM = compacted, NC = uncompacted, TGW = thousand-grain weight, stand = population of plants, Num pod = total number of pods, BD = soil bulk density, TP = total porosity, FCTP = water storage capacity, SACTP = air storage capacity (0–0.2 m soil depth).

## 4. Discussion

Overall, the improvement in PR caused by the use of millet contributes to favoring the root growth of the crop in the following season (i.e., soybean). This reduces the risk of yield loss caused by water stress [26], which is commonly reported in soybean-producing regions of Brazil. In contrast, maize has a less developed and aggressive root system compared to sorghum and millet, in addition to having a lower tolerance for water stress [12]. This makes the crop more susceptible to the harmful effects of soil compaction, which is reflected in higher PR values.

The *Cerrado* region of Brazil has an off-season with high temperatures during the months of May to September and a dry season, when rainfall rarely occurs and severe water deficits are common [13]. In this scenario, the harmful effects of soil compaction are enhanced, and the millet crop has proven to be an excellent alternative to minimize the problems caused by compaction, in addition to being a crop that has a high C:N ratio that contributes to the slow decomposition and mineralization of straw compared to other plants. This results in a gradual release of nutrients, benefiting the crop in succession [27].

In the 0–0.1 m soil layer, the soil bulk density was higher in the compacted soil condition, which led to the degradation of the soil physical conditions for plant roots [28]. However, soil macroporosity, microporosity, total porosity, and water and air storage capacity were not influenced by soil compaction. This absence of differences may be related to soil resilience during the second season of cultivation due to the action of soil wetting and drying cycles, which can contribute to the relief of soil compaction, even within one agricultural season [29,30].

Maize and sorghum provided soil water storage capacity values above 0.88 (much higher than the ideal value of 0.66 proposed by Reynolds et al. [21]) in the 0–0.1 m soil layer, indicating that the soil may be subject to aeration restrictions for microbial and root respiration under longer wet periods [31]. Thus, soil moisture conditions below the field capacity would be necessary for an adequate supply of oxygen to the root zone. Contrarily, the use of millet presented the value closest to the ideal condition. These results suggest that the use of millet as a second-season crop improved the balance between air and water in the soil and that this crop can be used as a compaction management strategy in the superficial layers of soils under NT.

In the 0.1–0.2 m soil layer, the use of millet as a second crop again had better soil physical properties than the other two second crops. These results suggest that millet cultivation contributes to the improvement in the soil physical environment, especially in soils under NT, where the compacted layers are located in the 0.07–0.20 m soil depth

layers [32]. It is important to emphasize that the crop root systems inserted in production systems are important strategies in soil physical improvement [33], and the more vigorous the root system, the greater the root growth and effectiveness in soil physical improvement. These results corroborate Torres et al. [34] and Silva et al. [35].

Among the agronomic characteristics evaluated in soybeans, the final population had no effect from the soil compaction or second crop. Despite the soybean showing high phenotypic plasticity, presenting the ability to compensate for eventual failures in the stand [36], maintaining the plant population is a fundamental agricultural practice to ensure potential yields. The effect of predecessor crops impacting the final population of soybean plants was not expected with the accomplishment of the present work. However, regarding the soil physical condition, a reduction in soybean seed emergence has been reported previously [31,37,38]. It is possible that in the present work failures in the soybean stand were not visualized due to the greater soil bulk density in the compacted soil. This condition was not able to provide a reduction in soybean emergence.

For plant height, the benefit of millet before soybean sowing was evidenced in comparison with maize and sorghum. Soybean plants had the greatest height in areas with previous millet cultivation. Furthermore, the treatment with millet cultivation provided similar height values of soybean plants between soil compaction treatments. This was linked to the improvements in the soil physical properties imposed by the millet root system, a fact that allowed a greater exploration of the rhizosphere environment by the soybean plants, facilitating access to water and nutrients. Higher soybean growth rates are seen when there are no water or nutritional restrictions for the plants [39].

For the soybean yield components, such as the number of pods per plant, the benefit of millet was again observed, a fact that can be explained by the improvements in the soil physical conditions, as previously discussed. Millet, as a species originating from the African continent, presents morphophysiological adaptations to develop in conditions of low water availability [40]. In this sense, this plant species has a recognized efficiency in water use compared to other plants, in addition to having a very aggressive root system, which can reach great depths in the soil [41]. These characteristics improve the rhizosphere environment when millet is included in crop rotation systems, with benefits not only for soil properties but also for the yields of species in succession [13].

The benefits resulting from millet cultivation prior to soybean sowing, which were seen for the yield components, plant population, and number of pods, were not as evident for the thousand-grain weight. Because it is a character strongly influenced by the genetics of the cultivar [42], in general the agronomic practices carried out throughout the soybean crop cycle have little effect on the mass accumulation in grains.

For the soybean yield, the results corroborate the logic of the data obtained for the other response variables since the crop was subjected to development with soil physical restrictions, such as higher PR and BD and lower soil total porosity, macroporosity and water storage capacity, and aeration, especially in the 0.1–0.2 m layer, a lower crop yield was observed. In addition, the impacts of the poor soil physical condition were measured in the agronomic characteristics, a fact that demonstrates that soybean, when subjected to development in restrictive environments for root growth, presents morphological changes capable of indicating this unfavorable condition for the crop [7].

The principal component analysis showed a greater relationship between the compacted soil condition and the soil bulk density and water storage capacity. Beutler et al. [28] pointed out that a soybean crop grown in compacted soil has morphological changes in the roots due to the physical impediment, characterized by having higher soil bulk density. The responses of the soybean are consequences of several factors. The evaluated characteristics had a negative relationship with the water storage capacity and a positive relationship with the soil air storage capacity, demonstrating that the imbalances in these evaluations directly influenced the soil physical quality for crop development. The increase in the soil water storage capacity, imposed by a higher soil bulk density and lower soil aeration, reduces plant accessibility to water and nutrients, causing limitations in plant metabolism

and growth [9,26]. The negative effects of soil compaction could be seen in the negative relationship between the soil density and the plant height and thousand-grain weight.

Among the second-season crops, the use of millet differs from other treatments, independent of the soil condition, which shows the improvement provided to the system by the millet (fasciculated) root system. Millet directly influences the decompaction of the soil, reducing the BD [27,28].

The results of this study confirmed that soil compaction caused by machine traffic reduced the soil physical quality, a fact that negatively impacted soybean agronomic responses. Thus, plants were more susceptible to abiotic stresses, whether due to an excess or a lack of water, causing damage to the agricultural production system. Therefore, the choice of second-season crops with greater tolerance to water stress and compaction by soybean cultivars from a higher maturation group, which in the case of this study was millet, can be an alternative to minimize the negative effects of soil compaction. Despite this, it is necessary to further improve the diversity of crops within the production system, prioritizing species that will contribute to improving the physical quality of the soil in a compacted soil environment. Thus, reductions in crop productivity can be mitigated, in addition to the recovery of soil structure.

## 5. Conclusions

Compaction influenced the soil physical quality, mainly in the 0.1–0.2 m layer, reflecting a decrease in the performance of the soybean crop.

The use of millet improved the soil physical properties, with an attenuation of penetration resistance of up to 20% compared to other crops.

Soybeans produced about 10% more after cultivation in succession to millet compared to maize and sorghum.

The choice of the predecessor crop for the cultivation of soybean impacts the soil physical properties, which may help in attenuating soil compaction as well as increasing soybean yield.

**Author Contributions:** Conceptualization, C.J.B.F. and A.G.d.S.; methodology, C.J.B.F., A.G.d.S. and C.A.T.; validation, C.J.B.F., V.R.d.O.P. and G.B.P.B.; formal analysis, C.J.B.F., C.A.T. and M.d.F.S.; investigation, C.J.B.F., A.G.d.S., V.R.d.O.P. and G.B.P.B.; data curation, C.J.B.F. and M.d.F.S.; writing—original draft preparation, C.J.B.F., G.B.P.B. and A.L.B.R.d.S.; writing—review and editing, all authors; visualization, C.J.B.F. and A.L.B.R.d.S.; supervision, C.J.B.F. and G.B.P.B.; funding acquisition, C.J.B.F. and A.G.d.S. All authors have read and agreed to the published version of the manuscript.

**Funding:** This research received external funding from Fundação de Amparo à Pesquisa do Estado de Goiás (FAPEG) projects 201810267001505 and 201810267001546, and the APC was funded by Universidade de Rio Verde (UniRV).

**Data Availability Statement:** The data are available from the corresponding author.

**Acknowledgments:** The authors would like to thank Fundação de Amparo à Pesquisa do Estado de Goiás (FAPEG) for research suport (Process numbers: 201810267001505 and 201810267001546).

**Conflicts of Interest:** The authors declare that this research was conducted in the absence of any commercial or financial relationships that could be construed as potential conflicts of interest.

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
