# Peer review of "Effects of Second-Season Crops on Soybean Cultivation in Compacted Soil in Brazilian Cerrado"

_agronomy, doi:10.3390/agronomy13010079_

Round 1

Reviewer 1 Report

Abstract: In the abstract of the works, the compaction of the soil is evaluated and some data (values) must be entered. At the same time, it would be advisable to write values ​​of the physical properties of the soil.

Material and Methods: A short description of the biological material used in the experiment. It would be advisable to write the soybean emergence (date).

Results: The Figure 3 (Analysis of the main components of soil conditions) can be a little bigger?

Conclusions: A general conclusion regarding the interaction of experimental factors on soil and yield.

Author Response

Dear Editor,

Follows a letter that addresses the new revisions made about the evaluation of the manuscript entitle "Effects of Second Season Crops on Soybean Cultivation in Compacted Soil in Brazilian Cerrado" which was submitted to "Agronomy" for the special issue “Tillage, Soil Management, and Field Traffic: Impact on Soil Physical and Mechanical Properties”. All the comments addressed be the Reviewers were considered and the justifications were presented point-by-point in attached.

Please do not hesitate to contact me for further information. Thanks in advance for your time and consideration.

Reviewer 2 Report

Dear Authors,

Congratulation for the good work.

Some notes:

1. Line 15 The used experimental design instead of The experimental design used

2. in Materials and Methods please numerized the subsection

3. Figure 2 and Table 2 change the space between the lines

Best Regards

Katerina Molla

Author Response

(The authors gave the same response as above.)

Author Response

(The authors gave the same response as above.)
